# High-Efficiency Carbon Fiber Recovery Method and Characterization of Carbon FIBER-Reinforced Epoxy/4,4′-Diaminodiphenyl Sulfone Composites

**DOI:** 10.3390/polym14235304

**Published:** 2022-12-04

**Authors:** Yong-Min Lee, Kwan-Woo Kim, Byung-Joo Kim

**Affiliations:** 1Convergence Research Division, Korea Carbon Industry Promotion Agency, Jeonju 54852, Republic of Korea; 2Department of Carbon Convergence and Composite Materials Engineering, Jeonbuk National University, Jeonju 54896, Republic of Korea; 3Department of Carbon-Nanomaterials Engineering, Jeonju University, Jeonju 55069, Republic of Korea

**Keywords:** carbon fiber-reinforced plastic (CFRP), recycling, epoxy, DDS, mechanical properties

## Abstract

Globally, the demand for carbon fiber-reinforced thermosetting plastics for various applications is increasing. As a result, the amount of waste from CFRPs is increasing every year, and the EU Council recommends recycling and reuse of CFRPs. Epoxy resin (EP) is used as a matrix for CFRPs, and amine hardeners are mainly used. However, no research has been conducted on recycling EP/4,4’-diaminodiphenyl sulfone (DDS)-based CFRP. In this study, the effect of steam and air pyrolysis conditions on the mechanical properties of re-cycled carbon fiber (r-CF) recovered from carbon fiber-reinforced thermosetting (epoxy/4,4′-diaminodiphenyl sulfone) plastics (CFRPs) was investigated. Steam pyrolysis enhanced resin degradation relative to N_2_. The tensile strength of the recovered r-CF was reduced by up to 35.12% due to oxidation by steam or air. However, the interfacial shear strength (IFSS) tended to increase by 9.18%, which is considered to be due to the increase in functional groups containing oxygen atoms and the roughness of the surface due to oxidation. The recycling of CFRP in both a steam and an air atmosphere caused a decrease in the tensile strength of r-CF. However, they were effective methods to recover r-CF that had a clean surface and increased IFSS.

## 1. Introduction

Carbon fiber-reinforced plastics (CFRPs) are high-performance composite materials composed of carbon fibers and thermosetting or thermoplastic polymer matrices [1,2,3]. A global goal is to achieve net-zero carbon dioxide emissions by 2050 [4]. Therefore, studies on improving fuel efficiency have been conducted in the fields of aerospace [5,6], ships [7,8], and automobiles [9,10] by building vehicle bodies and parts with CFRP, which is light, high intensity, and high elasticity for light weight [11,12,13]. As CFRP can replace metals due to the properties mentioned above, as well as the advantages of excellent corrosion resistance [14,15] and durability [16], demand has also increased in structural applications such as renewable energy [17,18], sports [19], and civil engineering and architecture [20,21,22]. As the production of and demand for CFRP increases, the amount of waste also increases yearly [23,24,25]. In the European Union, automobiles generate 8-9 million tons of waste each year at the end of their lifespan. Accordingly, the European Parliament and the Council adopted a directive on end-of-life vehicles in 1997 to address this issue. This directive requires that waste be reused and recovered. Thus, giving priority to reuse and recycling is the basic principle of the directive [26]. Meanwhile, delivery of approximately 40,000 aircraft is expected by 2038, with an annual air traffic growth rate of at least 4.3%. Of these aircraft, 56% are expected to accommodate market growth, and the remaining deliveries are expected to replace approximately 75% of current aircraft over the next 20 years. This will result in the retirement of approximately 19,000 commercial aircraft over the next 18 years. Due to regulations, OEMs consider the end of life of an aircraft at the design stage, and research on aircraft recycling is underway in industry and academia [27]. CFRP is used in 53% of the Airbus a350 XWB, and in 50% of the Boeing 787 [28]. The recent launch of clean aviation to achieve climate neutrality has driven the goal of replacing 75% of existing civil aircraft with hydrogen- and electric-powered aircraft [29]. By 2050, the aviation industry is expected to collect approximately 500,000 tons of end-of-life (EOL) waste from CFRP [30]. CFRP has been disposed of by landfill in the past. However, disposal by landfill is restricted or prohibited because it causes environmental problems due to the lifespan of polymers [31,32,33]. Therefore, various recycling methods for CFRP have recently been developed. There are three typical recycling methods for CFRP: mechanical, chemical, and thermal. Mechanical recycling is the simple process of physically crushing CFRP. However, CF and resin are not separated during the crushing, which is a drawback, and long fibers are broken during the grinding process, which results in poor physical properties [34]. The chemical method is a method that uses a solvent to decompose important chemical bonds of the used thermosetting resin. Acids [35,36,37], hydrogen peroxide [34,38], and supercritical fluids [39,40,41] are used as solvents. However, acids take a long time to decompose CFRP, and hydrogen peroxide is difficult to manage due to its high reactivity. In addition, although supercritical fluids have a high CF recovery rate and can recover CF with a clean surface and excellent physical properties, high temperatures and high-pressure conditions are required, and the cost of r-CF recovery is high. In addition, large-capacity and continuous processes are difficult, and environmental problems may occur due to processing with solvents. Conventional thermal recycling is a simple and continuous process that decomposes resin and recovers CF at a high temperature of approximately 500 °C or higher in N_2_ or air atmosphere. However, a large amount of energy is consumed, and undecomposed impurities generated during the process are adsorbed on the CF surface, leaving a residue. In addition, pyrolysis produces CO and CO_2_, which can lead to environmental problems [42]. To compensate for these shortcomings, Kim et al. suggested a decomposition method using superheated steam in 2017 [23]. Superheated steam can uniformly apply heat to CFRP and penetrate the resin to decompose rapidly. In addition, unlike other pyrolysis methods, it is possible to recover in the form of tar and thus generate less CO_2_. Epoxy is the most commonly used thermosetting resin in CFRP production, and amine curing agents are used primarily for epoxy curing. Among the commonly used 4,4′-diaminodiphenyl methane (DDM) and 4,4′-diaminodiphenyl sulfone (DDS), many studies have been conducted on recycling methods of EP/DDM-based CFRP, but EP/DDS-based CFRP recycling methods have not been studied. In this study, EP/DDS-based CFRPs were pyrolysis in steam and air atmosphere. The temperature conditions in each atmosphere were essential to remove the residual epoxy on the CF surface. Optimal process conditions such as atmosphere gas type, temperature, and holding time were derived, and the mechanical properties of the recovered r-CF were investigated.

## 2. Materials and Methods

For the CFRPs used in this study, polyacrylonitrile-based unidirectional carbon fiber (CF, T700, Toray, Tokyo, Japan) was used as a reinforcing material, and an amine curing agent (4,4′-diaminodiphenyl sulfone, DDS, Sigma-Aldrich, Saint Louis, MO, USA, AHEW of 62.08 g/eq, T_m_ of 172–175 °C) and a bifunctional epoxy resin (EP, diglycidyl ether of bisphenol A, Kukdo Chem., Seoul, Republic of Korea, YD-128, EEW of 184–190 g/eq, a viscosity of 11,500–13,500 cps at 25 °C) were mixed in an equivalent ratio and then impregnated using the hand lay-up method. At this time, the impregnated EP was 35 ± 2 wt.%. Curing was performed at 190 °C (10 °C/min) for 1 h based on the differential scanning calorimeter (DSC) results shown in Figure 1. The structures of EP and DDS are shown in Figure 2.

### 2.1. CFRP Pyrolysis

A quartz tube furnace (length of 950 mm, inner diameter of 8 mm) was used for the CFRP pyrolysis. The fabricated CFRP was placed on an alumina crucible and loaded into a quartz tube. The heating steam was generated at 170 °C using triple-distilled water (5 cc/min) in a pre-heater. CFRP was heated to 500–600 °C at a rate of 10 °C/min and pyrolyzed for 180 min, followed by post-treatment at 500–600 °C for 10 min with air (O_2_-21% under N_2_; 200 cc/min). Finally, recycled carbon fiber (r-CF) was recovered after cooling to room temperature. The nomenclature for r-CF according to the pyrolysis conditions is shown in Table 1.

### 2.2. Fixed Pyrolysis Conditions

The curing temperature of the EP/DDS mixture used as a matrix was measured using a differential scanning calorimeter (DSC, DSC 3, Mettler Toledo, Columbus, OH, USA). Approximately 7 mg of the mixture was placed in a 100 μL aluminum pan and heated from 30 °C to 300 °C at a rate of 10 °C/min under an N_2_ atmosphere (50 cc/min).

The thermal decomposition temperatures of the CF and EP were measured using a simultaneous thermal analyzer (STA, TGA/DSC 3+, Mettler Toledo). Approximately 4 mg of the samples were placed in a 70 μL aluminum pan and heated from room temperature to 500, 550, and 600 °C at a rate of 10 °C/min under an N_2_ and air atmosphere (50 cc/min).

### 2.3. Physical Properties

The tensile properties of a single fiber of Ar-CF and r-CF were measured using a universal testing machine (UTM, Seron, Seoul, Republic of Korea) according to ASTM D3379. At this time, the gauge length of the CF was set to 10 mm, and the draw-off clamp speed was set to 1 mm/min. After placing the CF between the grips of the UTM, both sides of the sample window were cut at the midpoint of the gauge. For each sample, at least 50 CFs were measured; a schematic diagram of the CF tensile strength sampling is shown in Figure 3a. In the tensile test, an error may occur if the carbon fiber is placed diagonally rather than in the center of the window.

The interfacial shear strength (IFSS) between the Ar-CF and r-CF and the EP resin was also measured using UTM. A single fiber was placed at the center of a paper window and adhered with an EP bond, and EP droplets with a diameter of approximately 100–200 µm were placed using a sharp needle. The EP droplet is a difunctional epoxy (EP, diglycidyl ether of bisphenol A, Kukdo Chem., YD-128, EEW of 184–190 g/eq, a viscosity of 11,500–13,500 cps at 25 °C) and an amine curing agent (4,4′-diaminodiphenyl methane, Sigma-Aldrich, Saint Louis, MO, USA, AHEW of 49.565 g/eq, Tm of 89 °C) was used. It was mixed for 20 min on a hot plate at 80 °C, and the temperature was maintained during the droplet operation. EP droplets were cured at 150 °C (10 °C/min) for 1 h. At least 25 CFs were measured for each sample, and the schematic diagram for the IFSS sampling of CFs is shown in Figure 3b. The IFSS test can be considered a failure if the carbon fiber is located diagonally to the window or if the epoxy bond is not fully cured and the carbon fiber rather than the EP droplet is pulled out.

### 2.4. Component Analysis

Functional groups on the Ar-CF and r-CF surfaces were detected in the 400–4000 cm^−1^ range using Fourier-transform infrared spectroscopy (FT-IR, Nicolet iS10, Thermo Scientific, Waltham, MA, USA). FT-IR samples were prepared as discs by grinding CF and potassium bromide (KBr, Sigma-Aldrich, Saint Louis, MO, USA) together and applying a clamp force of 7 tons for 2 min using a hydraulic press (CrushIR, PIKE Technologies, Madison, Wisconsin, WI, USA).

The surface properties of Ar-CF and r-CF were observed using field emission scanning electron microscopy (FE-SEM, S-4800, Hitachi, Tokyo, Japan). Ar-CF and r-CF were fixed in the SEM sample holder using carbon tape, and all images were obtained at 10^4^ magnifications at 1.0 × 10^−5^ torr and an acceleration voltage of 15 kV.

## 3. Results and Discussion

### 3.1. Pyrolysis Conditions of EP/DDS and Carbon Fiber

CF and cured EP/DDS thermal decomposition behaviors were measured by STA under a gas (N_2_ or air) atmosphere, and the thermal decomposition behavior under a steam atmosphere was measured by a self-made TGA furnace. The results are shown in Figure 4, and a schematic diagram is shown in Figure 5. The mass of CF was decreased by 2.81% (N_2_), 4.58% (steam), and 7.62% (air) at 600 °C. In addition, EP showed a mass reduction of 71.09% (N_2_), 85.40% (steam), and 96.64% (air) at 600 °C. The mass loss was the least under N_2_ atmosphere and the degree of damage to the CF was therefore also small. On the other hand, the degree of damage to CF was large under the air atmosphere because the mass loss was the highest. Under the steam atmosphere, the decrease in mass was between that of N_2_ and air. Even though the amount of EP char remaining after treatment showed a slight difference (approximately 14.31%) between N_2_ and steam, it exerted a significant influence during the post-treatment (air) time. The post-treatment can have a significant impact on CF damage. In conclusion, it is considered that the CF damage can be minimized by shortening the duration of air post-treatment because EP char can be quickly generated under steam.

### 3.2. Morphology of the r-CF

The scanning electron microscopy (SEM) images of the surface morphology of Ar-CF and r-CF are shown in Figure 6. In steam pyrolysis at 500–600 °C, the EP of CFRP did not completely decompose. Therefore, some EP remained on the CF surface. However, an increase in the pyrolysis temperature affected the EP decomposition. Then, air pyrolysis induced the complete removal of EP on the CF surface within a short time. However, air pyrolysis at 500 °C did not completely remove the EP char on the r-CF and remained in the form of crumbs. On the other hand, air pyrolysis at 550 °C completely removed the EP char on the r-CF surface, and the degree of etching on the surface was small. Air pyrolysis at 600 °C also completely removed the EP char on the CF surface. However, the r-CF surface was etched, and a curve was formed due to excessive air treatment. Based on the results above, the conditions of the 50-55-10 sample were effective for EP/DDS-based CFRP recycling.

### 3.3. Mechanical Properties of the r-CF

The tensile strength of the recovered r-CF is shown in Figure 7a. Equation (1) was used to calculate the tensile strength.
(1)Strength=F/A
where *F* is the force at the break of the fiber, and *A* is the average cross-sectional area of the fiber.

The tensile strength of r-CF was decreased as the pyrolysis temperature increased. In the case of Ar-CF, the tensile strength decreased from 4.67 GPa to 3.03 GPa (60-60-10). As the temperature increased to 500–600 °C, the tensile strength of r-CF tended toward 22.48% (4.49 to 3.62 GPa) in the case of the sample that only underwent steam pyrolysis. It is considered that the EP on the CF surface was decomposed by steam to form EP char, and thus the tensile strength was reduced as the CF was oxidized. The tensile strength of the 50-50-10 sample, which was subjected to air post-treatment after steam pyrolysis at 500 °C, was reduced by 11.99% (4.11 GPa), and the error rate was calculated to be large compared with that of the other samples. The EP char formed by steam was not completely decomposed in the air atmosphere and remained on the surface in the form of crumbs, which reduced the tensile strength and seemed to be the cause of the large error rate. The tensile strength of the sample subjected to secondary pyrolysis at 550–600 °C was reduced by up to 35.12% (3.85–3.03 GPa). Short air pyrolysis at temperatures above 550 °C induced a decrease in tensile strength due to oxidation and etching of the r-CF surface. However, the EP char on the CF surface was completely removed.

The IFSS of the recovered r-CF is shown in Figure 7b. Equation (2) was used to calculate the IFSS.
(2)IFSS=F/πDL
where *F* is the equipment force when the fiber breaks, *D* is the droplet size, and *L* is the diameter of the carbon fiber.

The IFSS of Ar-CF was 56.09 MPa. The IFSS of r-CF recovered under steam pyrolysis conditions was lower than the IFSS of Ar-CF, regardless of temperature conditions. However, the IFSS of r-CF after air treatment at 500–550 °C was calculated as 46.42 MPa (82.76%) and 49.95 MPa (89.05%), respectively, compared to Ar-CF. As steam-air pyrolysis proceeded, EP char debris was formed on the surface of r-CF, and EP droplets did not completely adhere to CF but instead formed on the EP char. Meanwhile, the IFSS of the r-CF after air treatment at 600° C for 10 min was calculated as 59.14 MPa (105.44%), 59.45 MPa (105.99%), and 61.24 MPa (109.18%), respectively, compared with Ar-CF. As shown in the SEM images in Figure 5, the specific surface area increased due to the curvature of the r-CF surface. In addition, it is thought that the improvement in the IFSS was caused by the increase in functional groups containing oxygen atoms on r-CF by air.

### 3.4. Functional Group of r-CF

Figure 8 shows the FT-IR analysis that was done to identify the change in functional groups containing oxygen atoms in Ar-CF and r-CF. The hydroxyl (-OH) group on the CF was observed at 3431 cm^−1^ [43,44]. The peaks corresponding to the sp^2^ C-H bond (=CH) and sp^3^ C-H bond (-CH) were observed at 2920 cm^−1^ and 2851 cm^−1^ [43,44,45], respectively. In addition, the peak for the C=C bond was observed at 1575 cm^−1^ [43,44,45], and the peaks for the ketone (C=O) and C-O bond were observed at 1638 cm^−1^ and 1100 cm^−1^ [43,44,45], respectively.

The FT-IR spectra of r-CF were similar to those of Ar-CF. As the steam-air heat treatment proceeded, the peaks corresponding to the -OH, C=O, and C-O groups increased, and the peaks corresponding to the -CH, =CH, and C=C bonds decreased, suggesting that steam and air heat treatment converted -CH, =CH, and C=C bonds to functional groups containing oxygen atoms. Therefore, it is considered that the introduction of functional groups containing oxygen atoms improved the interfacial bonding interaction between CF and EP.

## 4. Conclusions

In this study, CFRPs were prepared using epoxy (EP) and 4,4′-diaminodiphenyl sulfone (DDS) as a curing agent. The prepared CFRPs were pyrolyzed under steam and air atmospheres to recover r-CF, and the structural and mechanical properties of the r-CF were evaluated. As the pyrolysis temperature increased, r-CF was oxidized, and the tensile strength was reduced by up to 64.88% relative to that of Ar-CF. As the thermal decomposition temperature increased, r-CF was oxidized and the tensile strength decreased by up to 64.88% compared to Ar-CF, but the IFSS tended to increase by 9.18% due to the increase in oxygen functional groups. In conclusion, a short-duration air post-treatment after steam pyrolysis had the effect of the complete removal of the residual EP char on the CF surface. Based on these results, clean r-CF was recovered from EP/DDS-based CFRP after air post-treatment. Although the tensile strength decreased, the IFSS improved. Therefore, the r-CF with improved IFSS has advantageous for manufacturing high-quality CFRP because of its excellent interfacial bonding with EP.

## Figures and Tables

**Figure 1 polymers-14-05304-f001:**
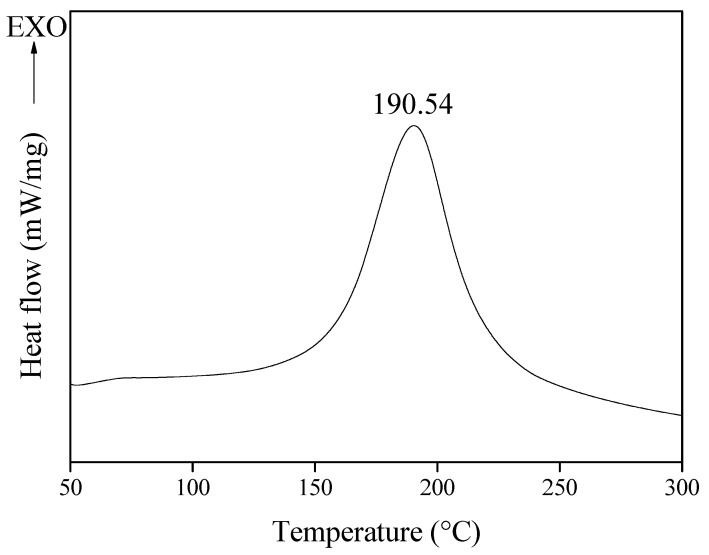
Curing temperature of epoxy/4,4′-diaminodiphenyl sulfone mixture using differential scanning calorimetry.

**Figure 2 polymers-14-05304-f002:**
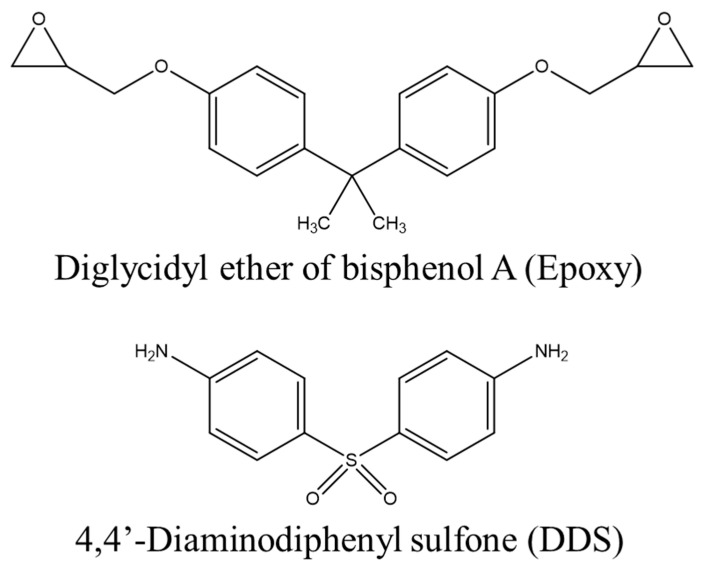
Chemical structures of epoxy and 4,4′-diaminodiphenyl sulfone.

**Figure 3 polymers-14-05304-f003:**
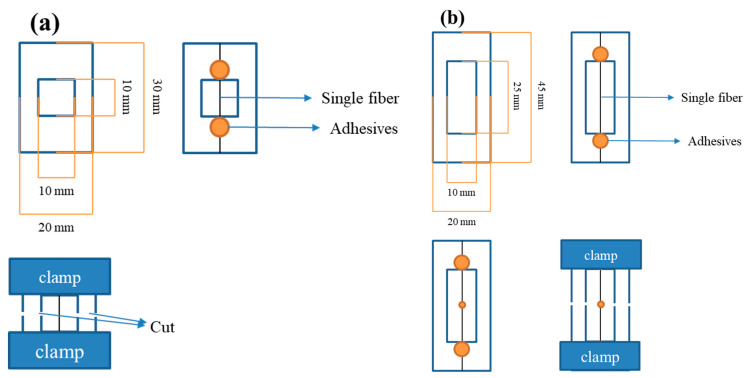
Schematics of physical properties test; (**a**) tensile strength and (**b**) interfacial shear strength.

**Figure 4 polymers-14-05304-f004:**
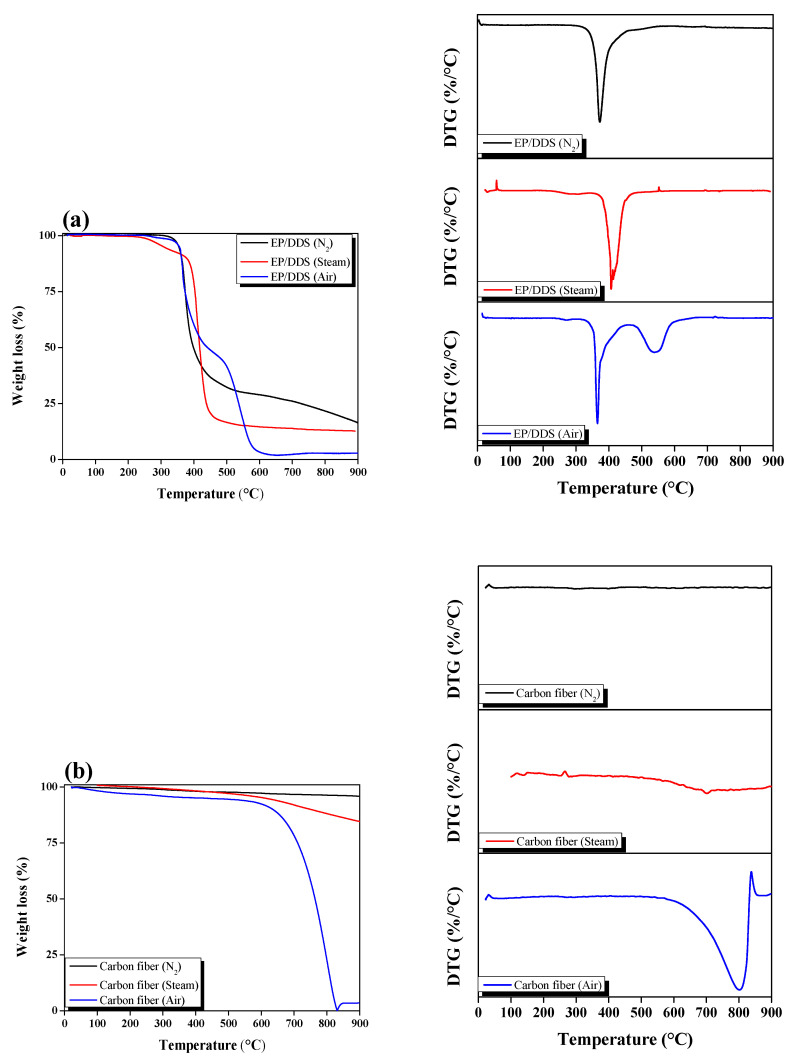
TGA and DTG graphs of as-received carbon fiber and epoxy resin pyrolysis under gas atmosphere: (**a**) cured epoxy/4,4′-diaminodiphenyl sulfone and (**b**) carbon fiber.

**Figure 5 polymers-14-05304-f005:**
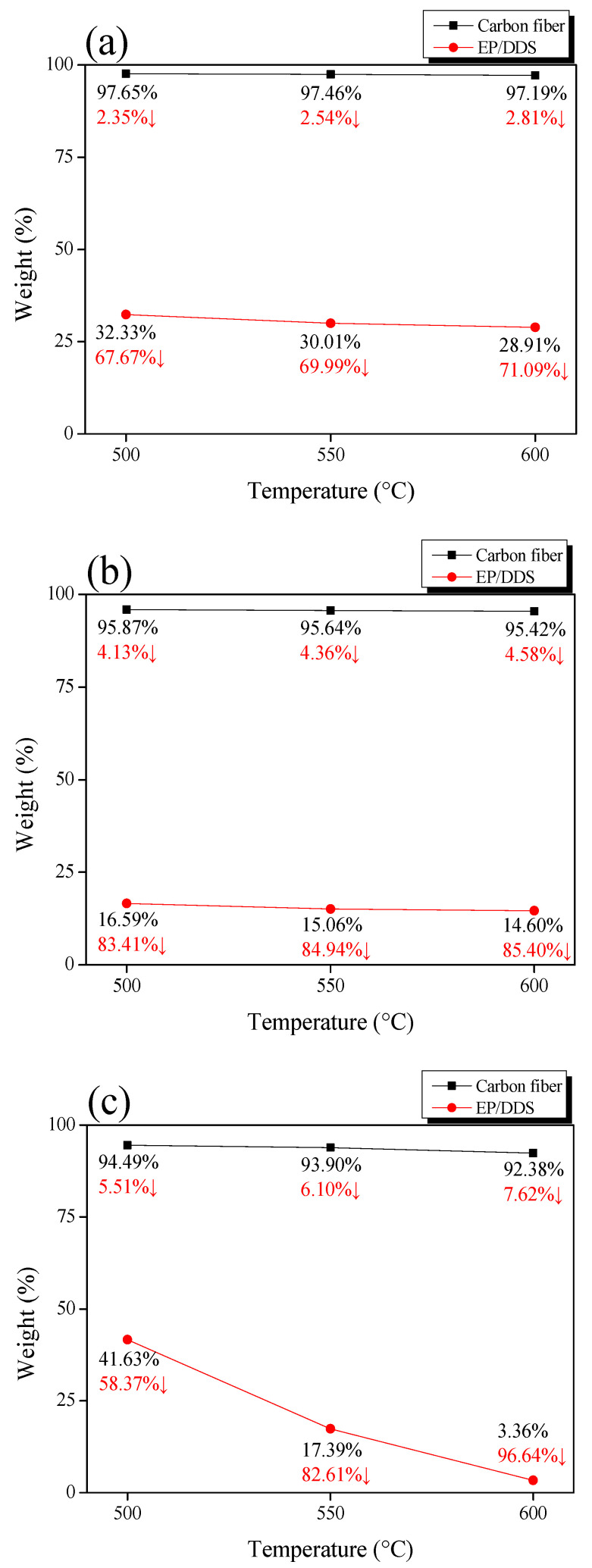
Weight loss of as-received carbon fiber and epoxy resin pyrolysis under gas atmosphere: (**a**) N_2_, (**b**) steam, and (**c**) air.

**Figure 6 polymers-14-05304-f006:**
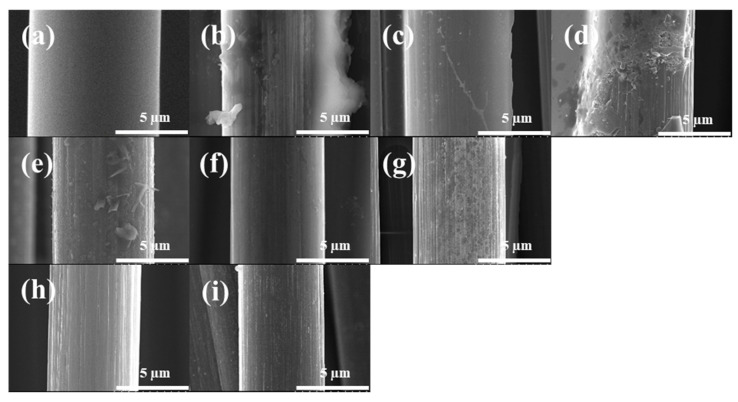
Scanning electron microscopy images of the as-received and recycled carbon fibers under varying pyrolysis conditions; (**a**) Ar-CF, (**b**) 50, (**c**) 55, (**d**) 60, (**e**) 50-50-10, (**f**) 50-55-10, (**g**) 50-60-10, (**h**) 55-60-10, and (**i**) 60-60-10.

**Figure 7 polymers-14-05304-f007:**
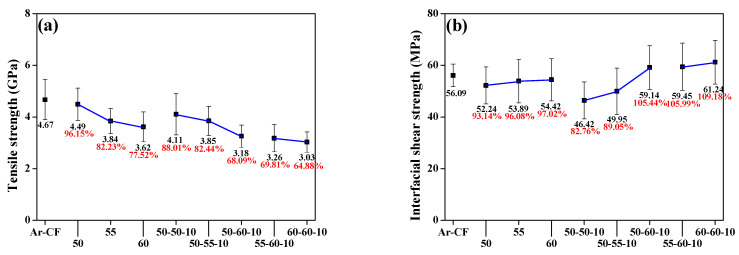
Physical properties of the as-received and recycled carbon fibers under varying pyrolysis conditions: (**a**) tensile strength and (**b**) interfacial shear strength.

**Figure 8 polymers-14-05304-f008:**
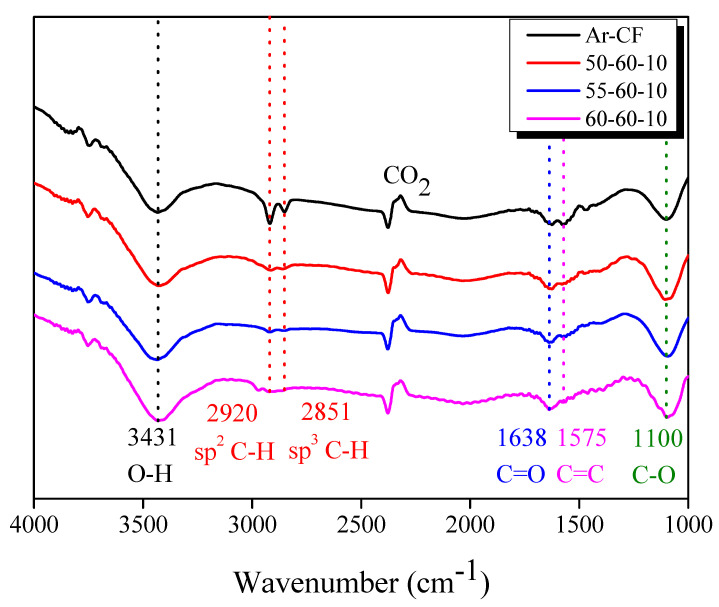
Fourier-transform infrared spectra of the as-received and recycled carbon fibers with varying pyrolysis conditions.

**Table 1 polymers-14-05304-t001:** Nomenclature for recycled carbon fibers and carbon fiber-reinforced plastics pyrolysis conditions.

Sample	Step 1				Step 2			
	Temperature (°C)	Heating Rate (°C/min)	Hold Time (min)	Steam (cc/min)	Temperature (°C)	Heating Rate (°C/min)	Hold Time (min)	Air (cc/min)
Ar-CF	-	-	-	-	-	-	-	-
50	500	10	180	5	-	-	-	-
55	550							
60	600							
50-50-10	500	10	180	5	500	10	10	200
50-55-10					550			
50-60-10					600			
55-60-10	550	10	180	5	600	10	10	200
60-60-10	600							

## Data Availability

The data presented in this study are available on request from the corresponding author.

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
