# Peer review of "High-Efficiency Carbon Fiber Recovery Method and Characterization of Carbon FIBER-Reinforced Epoxy/4,4′-Diaminodiphenyl Sulfone Composites"

_polymers, 2022, doi:10.3390/polym14235304_

Round 1
Reviewer 1 Report
This manuscript investigates the effect of steam and air pyrolysis conditions on the mechanical properties of recycled carbon fiber (r-CF) recovered from carbon fiber-reinforced thermosetting plastics. Tensile and interfacial shear tests were performed for the recovered r-CF. In a summary, conclusions in this study provide practical guidelines for the recycle of carbon fiber composites. I prefer to accept once the following issues addressed:
(1) Contents in the Highlights, Abstract, and Introduction need to be further improved to outstand the novelty and significance of this study.
(2) More details should be provided, such as test setup, failure images, and test curves for the carbon fiber.
(3) What are the novelties of this paper? Determine them and adjust the corresponding sections.
Reviewer 2 Report
This paper is a research on the recycled carbon fibers in composites through steam and heat treatment, and the tensile properties are decreased, but the interfacial properties are improved and analyzed. Due to the characteristics of recycled fibers, there is relatively improved IFSS despite their lower tensile properties through experiments. However, in the presentation and analysis of the results, the following areas need improvement.
Major
1. It seems that a more detailed explanation is needed in the IFSS measurement method. In the description of the test method, there is a difference between the epoxy droplet, however more accurate explanation is needed such as pulling method or fiber treatment. Since it is the most important data of this paper, it is considered that the definition of the experimental method is important.
2. In the sample manufacturing, the fiber fraction is described as about 65%. If it is correct, it seems that the relative loss of composites in the heat treatment process should be recorded. In the case of Figure 4, it seems that the thermal decomposition of carbon fiber and resin was recorded respectively, but in the case of composite samples, a detailed description of the results is needed.
3. In the IFSS measurement, whether the sample after heat treatment was subjected to the epoxy droplet mechanical test. It would be good to discuss in more detail about the treatment of residual epoxy after heat treatment. Also, if it seems necessary to measure in consideration of sample changes in the process, whether there is a change on the S-S curve or a difference in breakage. Since it is a test analysis that includes different remain materials and conditions, it is considered that there should be images or analysis of damage behavior.
4. In the IR analysis, it seems necessary to compare the difference between the as-received carbon fiber and the IR change of the carbon fiber that is not treated with epoxy. It seems that a comparative analysis is needed whether it is a change due to residual epoxy or a change to the carbon fiber surface or sizing material.
Minor
1. A tube furnace was used in the experimental equipment, and the tube furnace had a length of 950 mm and an inner diameter of 8 mm. please check the dimension once more.
2. It would be good to edit Table 1 more to make it look simpler.
3. Figure 5 looks like it needs realignment
4. It would be better that they have the same format for the amount of change in Figures 4 and 6.
Reviewer 3 Report
Kim et al studied influence of steam and air pyrolysis on mechanical properties such as IFSS or tensile strength of r-CF. The tensile strength increases while IFSS increases upon external treatment. Interesting results are provided and the topic is interesting. However, substantial revision is required before further consideration of the paper. Some points are –
[1] In abstract, please elaborate quantities obtained from experiments. Such as % of change in tensile strength or IFSS. The industrial applications needed to be summarized in abstract which are provide in introduction.
[2] A significant change in introduction is required. First of all, there are various sentences without references or support. Such as [a] “In the European Union, automobiles generate 8-9 million tons of waste each year”, [b] “delivery 38 of approximately 40,000 aircraft is expected by 2038, with an annual air traffic growth rate 39 of at least 4.3%.”, [c] “Of these aircraft, 56% are expected to accommodate market growth”, [d] “approximately 75% of current aircraft 41 over the next 20 years”, [e] “retirement of approximately 19,000 commercial aircraft over the next 18 years” and many more. These all should be referred or otherwise deleted.
[3] Please refer 4-5 related papers from MDPI journal and highlight the advancement of this work with existing work in MDPI in last paragraph of introduction.
[4] It is critical to state that authors never referred or validated the discussion and interpretation of experimental results especially the FTIR results should be supported with some references. In addition, there are total 42 references all of which are cited in introduction. What about rest of paper? Atleast some equations, text or validation of these results should be supported with references. Please fulfill this condition, otherwise paper will be rejected.
[5] In Figure 4, TGA and DTG curve should be presented. The values on Figure 4 can be summarized in form of a table.
[6] In Figure 5, scale bar must be visible. So, please make in bold and thicker.
[7] Conclusion must be improved. Please summarize the results and outcome of the experiments more concisely.
[8] English language must be improved with the help of native speaker.
Good Luck for revisions!
Round 2
Reviewer 3 Report
Minor revision requested -
[1] Please add TGA and DTG curves in the main text of the paper.
[2] In SEM images of Figure 5, the scale bar is not visible. Please make it bold and visible.
